# Observation of Nine Types of Spinach Pollen Morphology by Scanning Electron Microscopy

**Meng Meng** [†] , **Zhiyuan Liu** [†] **, Zhaosheng Xu, Helong Zhang, Hongbing She** * **and Wei Qian** *

State Key Laboratory of Vegetable Biobreeding, Institute of Vegetables and Flowers,
Chinese Academy of Agricultural Sciences, Beijing 100081, China; mm2129923954@163.com (M.M.);
liuzhiyuan01@caas.cn (Z.L.); xuzhaosheng@caas.cn (Z.X.); zhanghelong@caas.cn (H.Z.)
* Correspondence: shehongbing@caas.cn (H.S.); qianwei@caas.cn (W.Q.)
† These authors contributed equally to this work.

**Abstract:** Spinach (*Spinacia oleracea* L.) is a dioecious, wind-pollinated, and important green leafy vegetable that is widely cultivated worldwide. The characteristics of spinach pollen remain unclear. Herein, we investigated the pollen microscopic morphology of nine spinach genotypes by scanning electron microscopy (SEM). Because pollen grains of spinach are isodiametric and spheroidal, the following data for the pollen properties were observed: the maximum diameter of the pollen (27.66–33.05 um), pore diameter max (1.26–2.57 um), the number of visible pores (30–44), the density of pore elements/100 $\mu m^2$ (4.33–11.00), and the density of spinule elements/10 $\mu m^2$ (4.00–5.33). These are useful characteristics in distinguishing the species. Furthermore, by applying the cluster analysis method, based on key components, nine varieties are categorized into two main classes. These results provide a useful reference for the classification and identification of different spinach varieties.

**Keywords:** spinach; pollen features; scanning electron microscopy (SEM); classification





## 1. Introduction

Spinach (*Spinacia oleracea* L.), also referred to as Persian grass and red root vegetables, is an annual or biennial herb originating from Central Asia and has been cultivated for over 1300 years since its introduction to China in the 7th century AD [1]. According to FAO statistics, the total global production of spinach reached 30.1 million tons, of which China topped the list with 27.52 million tons, which accounts for 91 percent of the global production, followed by the United States and Turkey [2]. The green leaves of spinach are the most edible and nutritious part, rich in vitamin C, vitamins K, and B6, folic acid, and minerals (such as Fe and Ca). Spinach has garnered widespread appreciation worldwide due to its numerous benefits. However, the complex genetic backgrounds and similar morphological characteristics of spinach varieties pose challenges for variety classification and identification.

At present, the identification of spinach classification mainly focuses on morphological classification [3,4]. This approach relies mainly on observing and comparing external features, such as the shape of leaves and the shape of seeds. Often, one distinguishes different spinach varieties by looking at the shape of the leaves that can be intuitively divided into round and pointed leaves. Similarly, by observing the collective state of these seeds, we can determine whether they belong to wild species (usually clustered together) or cultivated varieties (usually dispersed). Although intuitive and easy to operate, this classification method has limitations. It can only be classified based on surface features, and it is difficult to explore the kinship relationship and evolutionary path among spinach varieties. Moreover, genetic differences are often overlooked in morphological classification, and different genetic information may be hidden under their similar appearance, thus preventing a comprehensive understanding of the biological nature of different spinach varieties and their unique mechanisms of adaptation to the environment. In order to overcome

these limitations, researchers are gradually exploring the combination of molecular marker technology and other modern biotechnology means, in order to reveal the genetic diversity and phylogenetic relationship among spinach varieties at the molecular level. For example, the amplified fragment length polymorphism (AFLP) molecular marker technology [5] is widely used in the study of the genetic diversity of spinach breeding materials. Through the analysis of DNA variation, the precise evolutionary tree of spinach varieties is constructed to deepen the understanding of the origin and evolution, and provide a scientific basis for the cultivation, of new varieties of specific excellent traits.

Pollen is the male gametophyte of higher plants, carrying the genetic information from the male individuals [6]. As a key carrier for seed plant reproduction, pollen not only ensures the continuation of species but also represents an important manifestation of genetic diversity and adaptability in nature. The morphological characteristics of pollen mainly include size, shape, exine ornamentation, and aperture morphology [7]. The pollen grains of each plant species possess a multitude of distinguishing pollen characteristics. The study of pollen morphology is especially important for plant identification and taxonomic research. This can help to determine the relationships among taxa at various taxonomic levels [8,9]. Morphological pollen features are precisely regulated by a series of complex gene networks. These genes determine the pollen's shape, size, texture, and color. The combination and expression of these genes enable the pollen of different plant populations to have their own characteristics, which maintains consistency within the populations and reflects the uniqueness between populations. The morphological features of pollen show a high degree of stability, which is not only reflected between individuals of the same plant species but also maintained under different environmental conditions. These traits have high genetic conservation among different species, implying that even more distantly related plant species may have similarities in some morphological pollen traits. This genetic conservation is a manifestation of their adaptability to the environment in the process of biological evolution and the diversity in the plant kingdom. In addition, the shape of pollen grains, the structure of their outer walls, and the characteristics of their ornamentation contain a wealth of information, serving as tools to analyze interspecific relationships, systematic evolution, and classification identification [10–16].

Technology for identifying pollen morphology plays a pivotal role in fruit tree variety classification. Scanning electron microscopy was used to analyze the size, shape, and surface texture of pollen grains under a microscope, providing a scientific basis for the accurate classification of fruit tree varieties. The diversity in pollen morphology among species and varieties makes it a key indicator for identifying fruit tree types [17] and is widely used to identify various types of fruit trees. By observing and analyzing morphological pollen characteristics, tree species and varieties can effectively be distinguished and identified. For example, apples and pears belong to the Rosaceae family, but their pollen morphology is significantly different, and they can quickly and accurately be differentiated using these subtle differences [18]. With advancements in modern technology, the identification of pollen morphology has become more precise and efficient, promoting the research and utilization of fruit tree genetic resources.

The study of pollen morphology has been commonly used in species recognition, the exploration of kinship relationships, and studies on origin and evolution. Since the morphological characteristics of pollen are relatively stable and contain genetic information, they are considered key factors in assessing species' evolutionary patterns and depth [19]. Due to its high resolution and image display, scanning electron microscopy is often used to observe morphological pollen characteristics [20]. Many plants are classified due to their unique pollen characteristics. The range is diverse, encompassing a variety of flowering plants [21,22] as well as multiple trees [23–25]. Since the 20th century, numerous scholars have conducted research on the pollen morphology of Chenopodiaceae plants [26,27]. But when we want to delve into the study of spinach, we find that research on its pollen morphology is quite scarce.

In this context, this study focused on the morphological characteristics of spinach pollen. Through a detailed morphological analysis, we aimed to reveal the structural features of spinach pollen and its potential significance in ecology and evolution. In the present study, scanning electron microscopy was used to examine the pollen grains of nine spinach varieties. Initially, precise measurements of their diameter, and other morphological characteristics were recorded. Subsequently, comparisons were made regarding differences in the number of visible pores on one face, pore densities, and number of spines on exine ornamentation. Cluster analysis was utilized to classify these varieties and identify potential genetic relationships among them. This research contributes to palynological evidence for the identification and systematic classification of plants within the genus *Spinacia*.

## 2. Materials and Methods

### 2.1. Plant Materials

Seeds from nine spinach varieties (Sp66, Sp46, Sp52, Sp134, Spl61, Sp71, Sp69, Sp05, and Sp136) were grown in the seed trays at the greenhouse of the Institute of Vegetables and Flowers, Chinese Academy of Agricultural Sciences (Beijing, China) during the spring of 2023. At the 3–4 leaf stage, plants were transplanted into pots. After 25 days of growth, spinach entered its flowering period. Pollen from male individuals for each variety was collected at 10 a.m., then soaked in FAA fixative (Sinopharm Chemical Reagent Co. Ltd., Shanghai, China) solution at 4 °C. To ensure the representativeness of the results, this study selected samples with significant characteristic differences. Specifically, the 9 pollen varieties collected included both cultivated and wild species. During sampling, pollen was extracted from 20 different plants for each genotype.

### 2.2. Morphology Studies

First, pollen was fixed in FAA containing 50% ethanol for 18 h and was rinsed three times with PBS buffer solution (Sinopharm Chemical Reagent Co. Ltd., Shanghai, China), each time for 10 min. Next, the pollen was dehydrated step-by-step through a gradient series of ethanol (Sinopharm Chemical Reagent Co. Ltd., Shanghai, China) concentrations (35%, 50%, 70%, 80%, 90%, 100%), with each step lasting for 15 min. Then, ethanol was replaced by tert-butyl alcohol (Sinopharm Chemical Reagent Co. Ltd., Shanghai, China) every 20 min, for a total of three replacements. Afterward, pollen was placed in a critical-point dryer (model: Quorum K850, Quorum Company, Edinburgh, UK) for drying [28]. Once dried, the sample was placed on a specimen holder with double-sided tape, and the samples were coated with a vacuum gold sputter using an ion sputter (model: Hitachi MC1000, Hitachi Hi-tech Company, Tokyo, Japan). Finally, pollen morphology was observed and photographed under a scanning electron microscope (model: HITACHI SU8100, Hitachi Company, Tokyo, Japan). It is important to emphasize that for scanning electron microscopy, dehydration, and drying techniques are of great importance. The principle of critical-point drying (CPD) is to avoid any damage to the pollen due to surface tension forces occurring during the transition from the liquid to the vapor phase. With critical-point drying, the pollen remains in its natural form under hydration, ensuring that the observations are closer to their true biological state. Each individual was randomly measured in 20 sets of sample data, and an average value was recorded. A broadly representative field of view was selected, and within the chosen observation area, 2K magnification was used to closely examine the individual morphology of the pollen grains. Using 18K magnification power, the exine ornamentation of the pollen grain and the structure of the aperture morphology were observed. These observations were captured, forming representative microscopic images. In summary, the morphological characteristics under study include diameter max (DM), area of one face, number of visible pores on one face (NoVP), pore diameter max (PDM), density of pore elements/100 $\mu m^2$ (DoP), and density of spinule elements/10 $\mu m^2$ (DoS). These representative images were recorded individually for further research and analysis. Because of this and natural variation, a

range categorizing pollen size is recommended: very small (<10 μm), small (10–25 μm), medium (26–50 μm), large (51–100 μm), and very large (>100 μm) [27,28]. Under previous classification methods, non-isopolar plant pollen will be graded according to P/E. It should be noted that because spinach pollen is isodiametric, without pollen polarity, and potentially deformed in the sediment, the polar axis length (P)/equatorial diameter (E) is impossible to estimate. So, what it shows in the text is the maximum diameter of the pollen measured.

### 2.3. Statistical Analysis

We conducted an in-depth statistical analysis of the data obtained from the experiment using Excel 2019 and SPSS 27.0. Firstly, an ANOVA was conducted on the data. Subsequent analyses using Duncan's multiple comparison tests should only be performed when significant differences between groups are confirmed. Duncan's new multiple-range test can be used for multiple comparisons to verify if there are significant differences among varieties, ensuring the accuracy of the results. Using SPSS 27.0 statistical software, comprehensive standardized processing was conducted on the collected pollen samples for the evaluated morphological characteristics. Inter-group linkage and Euclidean squared distance were used for cluster analysis to distinguish different pollen forms. Using this method, we determined significant differences among varieties, providing a crucial reference for subsequent research tasks.

## 3. Results and Analysis

### 3.1. Morphological Pollen Characteristics of Nine Spinach Varieties

Nine healthy spinach accessions were selected to observe the morphological characteristics of pollen (Figure 1). These accessions show significant morphological differences, including erect vs. semi-erect, serrated vs. round-shaped leaf, and light yellow vs. dark green leaf. Among nine accessions, Sp136 was wide-type, exhibiting narrow, hastate, and long petioles.

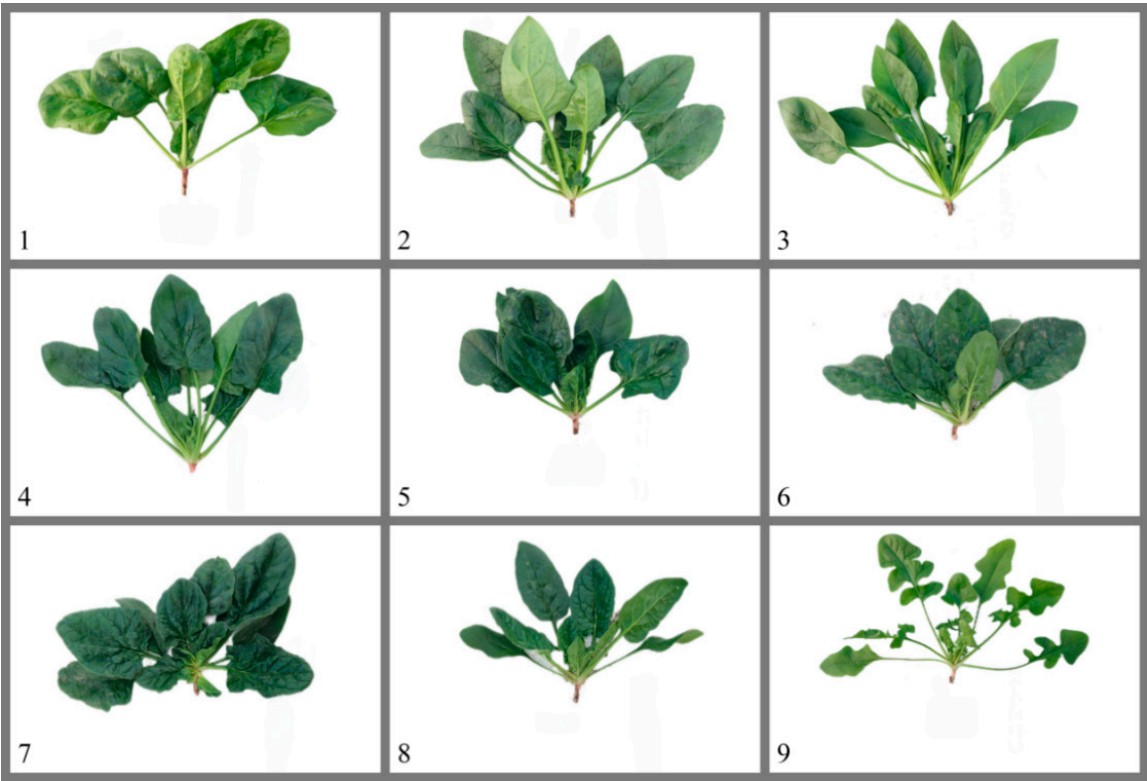

**Figure 1.** Characteristics of the nine spinach accessions: 1. Sp66; 2. Sp46; 3. Sp52; 4. Sp134; 5. Sp161; 6. Sp71; 7. Sp69; 8. Sp05; 9. Sp136.

The morphological characteristics of pollen showed marked differences (Table 1). Sp69 had a maximum diameter of 33.05 μm, which was significantly longer than any other variety ($p < 0.05$). At the same time, it had the largest single-plane area. Moreover, it had the largest number of pores visible in a single plane. While Sp46 had the greatest max pore diameter (2.57). Sp134 had the least max pore diameter (1.26). Sp71 contained the largest number of pores per 100 square micrometers (11.00).

**Table 1.** Morphological pollen characteristics of spinach varieties.

| ID | Sp66 | Sp46 | Sp52 | Sp134 | Sp161 | Sp71 | Sp69 | Sp05 | Sp136 |
|---|---|---|---|---|---|---|---|---|---|
| DM (um) | 30.34 (28.84–31.31) ± 1.07 | 30.67 (29.53–31.3) ± 0.81 | 30.78 (29.54–32.26) ± 1.12 | 27.66 (26.41–28.78) ± 0.97 | 31.96 (31.01–33.2) ± 0.92 | 31.60 (30.00–32.65) ± 1.15 | 33.05 (31.8–35.24) ± 1.55 | 28.64 (28.11–29.26) ± 0.47 | 30.07 (29.29–31.13) ± 0.78 |
| Area (um$^2$) | 820.5 (705.33–890.03) ± 82.02 | 836.11 (721.74–920.48) ± 83.86 | 832.15 (739.05–902.17) ± 68.56 | 653.02 (591.96–733.96) ± 59.65 | 852.65 (758.36–982.42) ± 94.85 | 849.05 (748.16–964.82) ± 89.07 | 928.72 (822.35–1021.44) ± 81.85 | 720.34 (623.1–802.27) ± 73.94 | 765.5 (689.83–878.31) ± 81.30 |
| NoVP | 40 (39–41) ± 0.5 | 37 (33–40) ± 2.5 | 30 (28–32) ± 1.0 | 37 (36–38) ± 0.82 | 42 (39–47) ± 3.56 | 31 (29–34) ± 2.16 | 44 (43–45) ± 0.82 | 32 (29–34) ± 2.16 | 39 (37–41) ± 1.63 |
| PDM (um) | 2.15 (2.05–2.24) ± 0.08 | 2.57 (2.28–2.99) ± 0.31 | 2.49 (2.14–2.94) ± 0.34 | 1.26 (1.1–1.47) ± 0.16 | 2.09 (1.95–2.22) ± 0.11 | 1.90 (1.84–1.98) ± 0.06 | 2.14 (2.12–2.17) ± 0.02 | 1.96 (1.73–2.3) ± 0.25 | 2.06 (1.79–22) ± 0.19 |
| DoP | 8.33 (8–9) ± 0.47 | 5.67 (5–6) ± 0.47 | 5.33 (4–7) ± 1.25 | 7.00 (6–8) ± 0.82 | 8.33 (8–9) ± 0.47 | 11 (8–15) ± 2.94 | 4.33 (3–6) ± 1.25 | 8.33 (7–10) ± 1.25 | 7.33 (7–8) ± 0.47 |
| DoS | 4.67 (4–5) ± 0.47 | 4.00 (4) ± 0 | 4.00 (4) ± 0 | 5.33 (5–6) ± 0.47 | 5.00 (4–6) ± 0.82 | 4.33 (4–5) ± 0.47 | 4.33 (4–5) ± 0.47 | 5.33 (5–6) ± 0.47 | 5.33 (5–6) ± 0.47 |

DM (Diameter Max), Area (Area of one face), NoVP (Number of Visible Pores on one face), PDM (Pore Diameter Max), DoP (Density of Pore, elements/100 μm$^2$), and DoS (Density of Spicule, elements/10 μm$^2$).

The pollen of these nine varieties was nearly spherical in shape (Table 1). According to the standards set by Halbritter et al. [29] for the classification of pollen size, diameters ranging from 26 to 50 μm are considered medium-sized pollen. Based on these size categories, the pollen shape of the nine spinach varieties was classified as medium-sized. The regions marked in Figure 2a,b show the germination pores along with sculpture elements (microechinate-perforate) (Figure 2). In this study, the pollen surfaces of all nine spinach varieties exhibited nanoechinate, with no significant differences in surface texture. However, characteristics such as DM, Area, NoVP, PDM, DoP, and DoS varied significantly among spinach varieties. These features can serve as auxiliary references for identifying different categories.

### 3.2. Pollen Morphological Feature Cluster Analysis

The results of standardizing the aforementioned data and applying the average linkage method (within-group) for classification research are shown (Figure 3). Based on a Euclidean distance of 15 as the cut-off point, 9 spinach varieties were classified into 2 categories. The first category included seven varieties: Sp46, Sp52, Sp66, Sp161, Sp71, and Sp69. The second category included three varieties: Sp05, Sp136, and Sp134. The kinship among breeds can be clearly judged by cluster analysis.

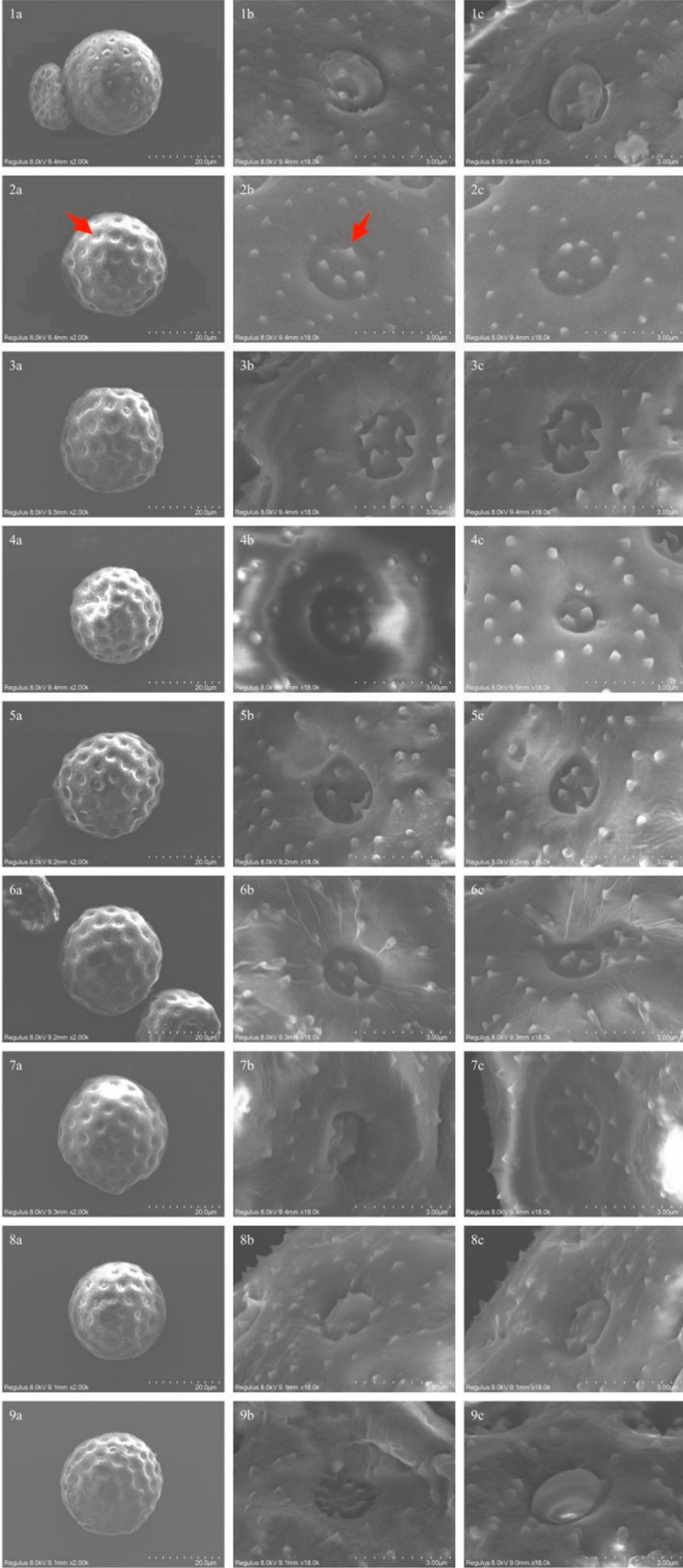

**Figure 2.** SEM photos of pollen grain morphology of nine spinach varieties: (**a**) Pollen morphology; (**b**) Germinal pore; (**c**) Exine ornamentation: (**1a–1c**) Sp66; (**2a–2c**) Sp46; (**3a–3c**) Sp52; (**4a–4c**) Sp134; (**5a–5c**) Sp161; (**6a–6c**) Sp71; (**7a–7c**) Sp69; (**8a–8c**) Sp05; (**9a–9c**) Sp136. The regions marked in (**2a,b**) show the germination pores along with sculpture elements (microechinate-perforate).

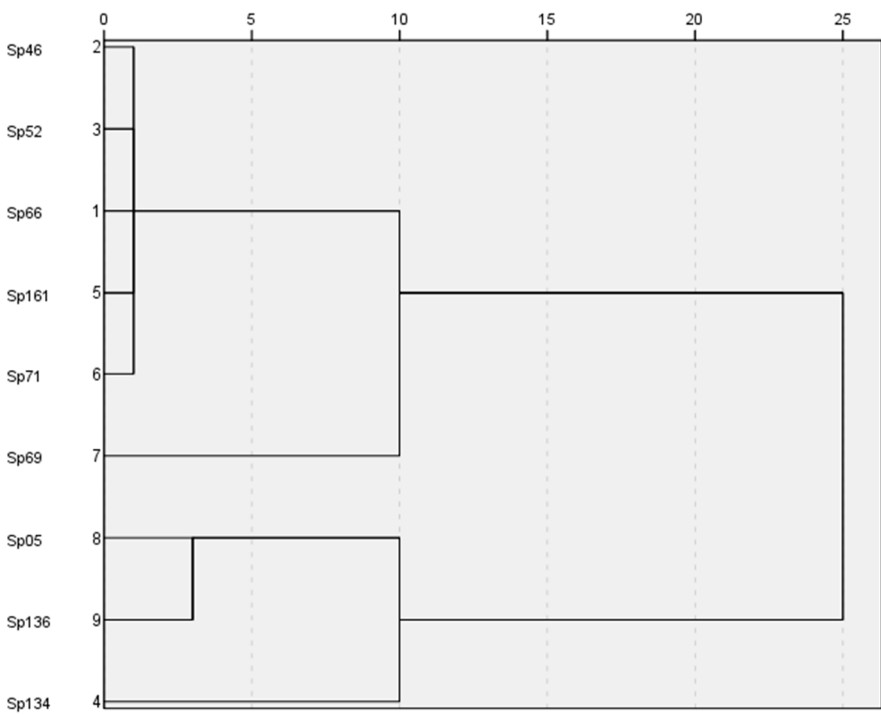

**Figure 3.** Cluster analysis of pollen morphology of 9 spinach cultivars.

## 4. Discussion

In botanical research, the classification of plants is a fundamental and significant work. Relying solely on the external morphology of plants can only yield a preliminary and intuitive impression. To obtain a deeper and more precise assessment, combining morphological classification with molecular marker techniques is supported. Pollen is a key element of plant reproduction, and in the process of evolution, the characteristics of pollen are strictly controlled by genetic factors and are usually not easily affected by environmental changes, exhibiting remarkable stability and specificity. This characteristic makes pollen an important resource for studying the evolutionary history and relationships of plants [30]. Consequently, the morphological traits of pollen play a crucial role in the identification and classification processes, providing additional tools for further exploration in plant taxonomy [31].

The morphological observation of pollen has been used to classify and uncover novel structures and functions in several species. Serisha et al. [32] employed advanced scanning electron microscopy technology to observe the pollen grains of *Artemisia annua*, revealing their unique morphological features and providing new insights into their biological structure and function. Yang et al. [33] conducted a comprehensive analysis of the microstructures of rice stems and leaves for the first time, promoting a deeper understanding of Davidian classification and palynology characteristics. Rashid et al. [34] utilized scanning electron microscopy to explore the characteristics of the pollen grain walls of *Astragalus* species, inferring that changes in pollen features can be stably maintained at the genetic level.

In this study, we employed advanced scanning electron microscopy technology to observe and analyze nine spinach varieties. This process not only revealed the unique structural features of spinach pollen at the microscopic level but also enhanced our understanding of the biological characteristics. Before conducting data analysis, we tried to visually distinguish nine pollen species from SEM images. However, after examining all pollen grains pictures, we concluded that the differences between them could not be confidently distinguished by the human eye. Although these spinach varieties exhibit significant differences in plant appearance and color, particularly the wild type Sp136, their pollen size classification and shape demonstrate consistency. Specifically, all samples are

medium-sized and exhibit a nearly subspherical state. The majority of the varieties have well-distinguished echini, a feature consistently found in the Chenopodiaceae family [35]. Our results similarly confirm this property. This morphological similarity may reflect fundamental biological characteristics that are crucial to their reproduction and survival, which have been preserved throughout the long evolutionary history of spinach. Cluster analysis based on morphological pollen characteristics revealed significant differences among spinach varieties. Based on a Euclidean distance of 15, the 9 spinach varieties were categorized into 2 groups: a large group consisting of varieties Sp46, Sp52, Sp66, Sp161, Sp71, and Sp69, and a smaller group containing Sp05, Sp136, and Sp134.

This study reveals the genetic relationships and hierarchical information among different spinach varieties, providing basic data. However, due to the underutilization of quantitative characteristics of plant pollen morphology in cluster analysis, the conclusions are somewhat limited. To more accurately identify the genetic relationships and classification of spinach varieties, further in-depth research is needed. A comprehensive analysis should consider various factors related to botany, genetics, cytology, and molecular biology. Through interdisciplinary approaches, a deeper understanding of spinach varieties can be achieved from multiple perspectives, such as observing morphological features like leaf shape, color, and stem thickness for intuitive information; genetics to reveal genetic differences; cytology to understand cellular structure characteristics; and molecular biology to reveal molecular level differences. Only by combining these diverse research methods can more accurate and reliable data be provided to determine the classification and genetic relationships of spinach. Such comprehensive research will not only advance spinach breeding progress but also provide powerful references for scientific research in related fields.

### 5. Conclusions

The pollen morphology results, researched with SEM for the nine spinach genotypes, indicated that all samples are medium-sized and exhibit a nearly subspherical state, and all varieties have good spinules. The difference between them is mainly reflected in the following aspects: the maximum diameter of the pollen (27.66–33.05 um), pore diameter max (1.26–2.57 um), the number of visible pores on one face (30–44), the density of pore elements/100 $\mu m^2$ (4.33–11.00), and the density of spinule elements/10 $\mu m^2$ (4.00–5.33). The nine spinach genotypes were classified into two categories based on the cluster analysis of pollen characters. This study enriches our knowledge of palynology and provides reliable information for the classification of spinach genotypes.

**Author Contributions:** Conceptualization, W.Q. and H.S.; methodology, Z.L., H.S. and M.M.; software, Z.L. and M.M.; validation, Z.L., H.S. and M.M.; formal analysis, H.S. and M.M.; investigation, H.S.; resources, Z.X. and H.Z.; data curation, M.M.; writing—original draft preparation, M.M., H.S. and Z.L.; writing—review and editing, Z.L.; visualization, Z.L. and M.M.; supervision, Z.L. All authors have read and agreed to the published version of the manuscript.

**Funding:** This work was supported by National Key R&D Program of China (2023YFD1200102), China Agricultural Research System (CARS-23-A17), and Chinese Academy of Agricultural Sciences Innovation Project (CAAS-ASTIP-IVFCAAS).

**Data Availability Statement:** The original contributions presented in this study are included in the article. Further inquiries can be directed to the corresponding authors.

**Conflicts of Interest:** The authors declare no conflicts of interest.

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
