# Peer review of "Observation of Nine Types of Spinach Pollen Morphology by Scanning Electron Microscopy"

_horticulturae, doi:10.3390/horticulturae10121358_

Round 1
Reviewer 1 Report
Comments and Suggestions for Authors
Review of the manuscript entitled “Observation of Nine Types of Spinach Pollen Morphology by Scanning Electron Microscopy”
In this study the pollen morphology of nine spinach (Spinacia oleracea L.) genotypes is investigated using scanning electron microscopy (SEM). The results showed that the pollen grains were nearly spherical, with a size range between 733.93 and 1021.44 μm². Key measurements such as the polar and equatorial axis lengths varied across genotypes, and a pollen shape ratio (P/E) ranged from 0.95 to 1.07. Cluster and principal component analyses revealed that these morphological traits—pollen size, shape, and axis lengths—can be used to classify and differentiate spinach varieties. These findings are used for classification and identification of spinach genotypes and their relatives.
General Overview of Major Issues
The authors investigated the role played by pollen morphology in delimitation of Spinach genotypes and their relatives. Overall, the manuscript requires significant revisions before it can be considered for publication. Several aspects of the manuscript require substantial revision, particularly the methodology (pollen measurements), and the conclusions based of these results as well as their statistical analysis. The manuscript is in principle clearly written, but the English needs to be improved. Additionally, some sections of the manuscript lack coherence, such as the discussion, and need to be reorganized to improve the overall flow of ideas. Some of the cited references are related to classification of family, genus, species level, but not directly related to the classification of genotypes. Although pollen can be useful for systematics, it is generally rarely useful for intraspecific classification. It is quite difficult to distinguish different species within a genus, and only in a few cases the pollen grains of varieties of a species can be differentiated. Regarding the pollen descriptions, the authors should be consistent and use/cite a pollen terminology (Punt et al. 2007 and/or Halbritter et al. 2018). My main concern is the method used for SEM preparation (literature reference also missing) and especially the pollen size measurements taken (see also comments in the manuscript.pdf). The pollen size is known to vary even within a single anther, and the reason why this pollen characteristic must be handled with care and considered as less reliable than other pollen characteristics. Regarding the results of the presented pollen study and as seen on the illustrated SEM micrographs, it seems that the classification of spinach varieties is not possible based on their pollen morphology. Whereas a classification based on the leaf morphology as illustrated in Figure 1 seems more likely to me.
Specific suggestions
I have inserted comments directly in the manuscript, and I will summarize the most important changes that needs to be done:
Introduction
The authors mentioned literature where the study of pollen morphology is important/useful to determine taxa on family/genus/species level. The manuscript would benefit from a more thorough engagement with the current literature on intraspecific classification based on pollen morphology, such as e.g.,
Lechowicz K, Bocianowski J, Wrońska-Pilarek D. Pollen Morphological Inter- and Intraspecific Variability in Selected Species of Rubus L. (Rosaceae). Forests. 2022; 13(11):1946. https://doi.org/10.3390/f13111946
Moreover, there could be a more general overview and more information about the pollen characteristics typical for the family Amaranthaceae. Several pollen investigations were carried out on Amaranthaceae s.lat. (Alkhani & al. 2003; Iamonico 2009; McAndrews & Swanson 1967; Pinar & Inceoglu 1999; Riollet & Bonnefille 1976; Shepherd & al. 2005; Soliman 2006; Toderich & al. 2010; Tsukada 1967; Zare & Keshavarzi 2007). Typical for the family are pantoporate, echinate pollen, and the pollen morphology of most species is very similar. It might be useful to understand the problem of classification of different taxa within this family and their genera and species by pollen morphology.
Material and Methods
The methodology section requires further clarification, particularly in how the pollen material was prepared for critical point drying and regarding the conducted pollen measurements. As spinach pollen is spheroidal and isodiametric a P/E ratio cannot be measured, this is only possible in prolate and oblate pollen grains. As these results were used for statistical analysis and further interpretation and delimitation of the genotypes, the whole methodology needs to be revised. Moreover, a combined study with light and electron microscopy would also be useful. In LM the exine thickness can be measured, and pollen size measurements can be made on pollen in hydrated condition, instead of semi-hydrated pollen in SEM.
Please compare other papers on Amaranthaceae pollen studies for parameters measured or calculated, such as in the study by Angeli et al. (2014): diameter, volume, exine thickness, number of pores, pore density, interpore distance, pores size (P os: max diameter; E os: min diameter). The total number of pores for each grain, a homogeneous distribution of pores on the pollen grain surface and the hexagonal area surrounding each pore is considered.
Methods for measurements on Amaranthaceae pollen can be found in the respective literature, such as e.g.:
Angelini, P., Bricchi, E., Gigante, D., Poponessi, S., Spina, A. & Venanzoni, R.: Pollen morphology of some species of Amaranthaceae s. lat. common in Italy. — Fl. Medit. 24: 247-272. 2014.
Information on “how to measure pollen grains” are also on the PalDat Website (https://www.paldat.org/howto/howto).
Results
The results need revision based on relevant pollen measurements and accurate pollen descriptions. The quality of the critical point drying process and their interpretation is not sufficiently tied to the research questions. It would be helpful to include light microscopy and a more detailed description of the pollen morphology to be able to answer the research questions and how they relate to the existing literature. Pollen descriptions need to be revised (such as the interpretation of the exine ornamentation) and only standard pollen terms should be used (see pdf comments).
Discussion
The discussion needs to be reorganized to improve the overall flow of ideas and needs to be changed based on the revised results

The manuscript is in principle clearly written, but the English needs to be improved.
Author Response
# Response: Dear Reviewer, we are deeply grateful for your recognition of and invaluable suggestions regarding our manuscript. After carefully reviewing your feedback, it occurred to me that significant improvements were necessary in terms of linguistic fluency, detailed description of methodologies, depth of discussion, and accuracy of writing. Your guidance has been instrumental in enhancing the overall quality of this work. I have implemented the revisions as per your recommendations.
General Overview of Major Issues
- Overall, the manuscript requires significant revisions before it can be considered for publication. Several aspects of the manuscript require substantial revision, particularly the methodology (pollen measurements), and the conclusions based of these results as well as their statistical analysis.
# Response: Thank you very much for the valuable comments provided by the reviewer. The sections on methodology, statistical analysis, and discussion of results have been revised and elaborated more thoroughly.
- The manuscript is in principle clearly written, but the English needs to be improved.
# Response: Thanks for your suggestion. We have tried our best to polish the language in the revised manuscript.
- Additionally, some sections of the manuscript lack coherence, such as the discussion, and need to be reorganized to improve the overall flow of ideas.
# Response: We fully agree with the reviewer’s suggestions regarding the logical expression of the article. We read through the full text, reorganized the language and improved some of the content.
- Some of the cited references are related to classification of family, genus, species level, but not directly related to the classification of genotypes. Although pollen can be useful for systematics, it is generally rarely useful for intraspecific classification. It is quite difficult to distinguish different species within a genus, and only in a few cases the pollen grains of varieties of a species can be differentiated.
# Response: We agree with your perspective that most studies primarily emphasizes taxonomic studies at the species, genus, and family levels. In our study, including cultivated varieties and wild species , the purpose of study is to observe pollen morphology and measure morphological indicators to compare the similarities and differences of different varieties. Indeed ,there are slight difference among these accessions.
- Regarding the pollen descriptions, the authors should be consistent and use/cite a pollen terminology (Punt et al. 2007 and/or Halbritter et al. 2018).
# Response: The section on pollen descriptions has been re-written as reviewer's suggestions. After reading through the full text, the non-standard language has been modified into the pollen terminology.
- My main concern is the method used for SEM preparation (literature reference also missing) (see also comments in the manuscript.pdf).
# Response: Thank you very much for the reviewer's detailed comments and valuable suggestions on the article. The specific operation of pollen observation by SEM is as follows: First, Pollen was fixed in FAA containing 50% ethanol for 18 hours was rinsed three times with PBS buffer solution, each time for 10 minutes. Next, the pollen was dehydrated step-by-step through a gradient series of ethanol concentrations (35%, 50%, 70%, 80%, 90%, 100%), with each step lasting for 15 minutes. Then, ethanol was replaced by tert-butyl alcohol every 20 minutes, for a total of three replacements. Afterwards, pollen was placed in a critical point dryer (model: Quorum K850) for drying[28]. Once dried, place the sample on a specimen holder with double-sided tape, the samples were coated with a vacuum gold sputter using an ion sputter (model: Hitachi MC1000). Finally, pollen morphology was observed and photographed under a scanning electron microscope (model: HITACHI SU8100).
- The pollen size is known to vary even within a single anther, and the reason why this pollen characteristic must be handled with care and considered as less reliable than other pollen characteristics. Regarding the results of the presented pollen study and as seen on the illustrated SEM micrographs, it seems that the classification of spinach varieties is not possible based on their pollen morphology. Whereas a classification based on the leaf morphology as illustrated in Figure 1 seems more likely to me.
# Response: We sincerely appreciate the reviewer's valuable feedback. Although leaf morphology can be categorized and offers a more intuitive representation, it is significantly influenced by environmental factors. In contrast, pollen remains relatively stable. The results obtained from measuring nine varieties in this experiment indeed demonstrated disparities in pollen characteristic. Furthermore, extensive research indicates that pollen characteristic serves as an effective criterion for differentiating various plant categories. Therefore, pollen characteristic can also be considered a reliable classification indicator for spinach.
Specific suggestions
I have inserted comments directly in the manuscript, and I will summarize the most important changes that needs to be done:
Introduction
The authors mentioned literature where the study of pollen morphology is important/useful to determine taxa on family/genus/species level. The manuscript would benefit from a more thorough engagement with the current literature on intraspecific classification based on pollen morphology, such as e.g.,
Lechowicz K, Bocianowski J, Wrońska-Pilarek D. Pollen Morphological Inter- and Intraspecific Variability in Selected Species of Rubus L. (Rosaceae). Forests. 2022; 13(11):1946. https://doi.org/10.3390/f13111946
Moreover, there could be a more general overview and more information about the pollen characteristics typical for the family Amaranthaceae. Several pollen investigations were carried out on Amaranthaceae s.lat. (Alkhani & al. 2003; Iamonico 2009; McAndrews & Swanson 1967; Pinar & Inceoglu 1999; Riollet & Bonnefille 1976; Shepherd & al. 2005; Soliman 2006; Toderich & al. 2010; Tsukada 1967; Zare & Keshavarzi 2007). Typical for the family are pantoporate, echinate pollen, and the pollen morphology of most species is very similar. It might be useful to understand the problem of classification of different taxa within this family and their genera and species by pollen morphology.
# Response: We are grateful to the reviewers for providing numerous references related to spinach, and selectively citing them in our article.
Material and Methods
The methodology section requires further clarification, particularly in how the pollen material was prepared for critical point drying and regarding the conducted pollen measurements. As spinach pollen is spheroidal and isodiametric a P/E ratio cannot be measured, this is only possible in prolate and oblate pollen grains. As these results were used for statistical analysis and further interpretation and delimitation of the genotypes, the whole methodology needs to be revised. Moreover, a combined study with light and electron microscopy would also be useful. In LM the exine thickness can be measured, and pollen size measurements can be made on pollen in hydrated condition, instead of semi-hydrated pollen in SEM.
Please compare other papers on Amaranthaceae pollen studies for parameters measured or calculated, such as in the study by Angeli et al. (2014): diameter, volume, exine thickness, number of pores, pore density, interpore distance, pores size (P os: max diameter; E os: min diameter). The total number of pores for each grain, a homogeneous distribution of pores on the pollen grain surface and the hexagonal area surrounding each pore is considered.
Methods for measurements on Amaranthaceae pollen can be found in the respective literature, such as e.g.:
Angelini, P., Bricchi, E., Gigante, D., Poponessi, S., Spina, A. & Venanzoni, R.: Pollen morphology of some species of Amaranthaceae s. lat. common in Italy. — Fl. Medit. 24: 247-272. 2014.
Information on “how to measure pollen grains” are also on the PalDat Website (https://www.paldat.org/howto/howto).
# Response: This has been corrected in manuscript.
Results
The results need revision based on relevant pollen measurements and accurate pollen descriptions. The quality of the critical point drying process and their interpretation is not sufficiently tied to the research questions. It would be helpful to include light microscopy and a more detailed description of the pollen morphology to be able to answer the research questions and how they relate to the existing literature. Pollen descriptions need to be revised (such as the interpretation of the exine ornamentation) and only standard pollen terms should be used (see pdf comments).
# Response: This has been corrected in manuscript.
Discussion
The discussion needs to be reorganized to improve the overall flow of ideas and needs to be changed based on the revised results.
# Response: This has been corrected in manuscript.

Reviewer 2 Report
Comments and Suggestions for Authors
Dear authors
The study, represented in the manuscript, will try to research the pollen morphology of nine spinach varieties by using SEM. This work would be useful in drawing conclusions about the genetic and morphological variability in spinach and should be taken into consideration for purposes of classification and identification.
Although the manuscript discusses an interesting and useful topic, its results are related to taxonomy, plant breeding, and conservation. However, there are several drawbacks and aspects to be improved in relation to readability, clarity, and scientific thoroughness.
Strengths
Relevance and Contribution: This research covers huge lacuna in the literature on the morphology of spinach pollen, which has been comparatively unexplored until now. Valuable data about classification and further investigation of Spinacia oleracea were presented.
Methodology: Scanning electron microscopy is applied, which allows the observation of pollen grains at high resolution, giving one all the morphological details.
It helps in understanding the interlinkages between different spinach varieties. Cluster analysis and principal component analysis will help in the final data analysis.
Shortcomings and Mistakes
Abstract Clarity:
The abstract should be not so wordy. It should make the findings important more visible, and the abstract should summarize points and results arising rather than stating again the rationale of the study.
Typographical Errors:
There are a number of typographical errors throughout the manuscript; for example, "markly" should read "markedly," "plolar" should read "polar." The manuscript needs careful proofreading to eliminate such errors.
Structure and Flow:
The introduction is somewhat disjointed in terms of information flow. The authors need to provide background information in a more logical manner; for instance, starting off more generally with spinach before getting into pollen morphology.
Materials and Methods:
It is essential to add more specificity to the description of the sample collection, specifically mentioning how many plants represent each genotype when pollen was sampled. This is very important to provide reproducibility. Here, a detailed explanation of the fixation process for pollen is needed. More precisely, the concentration and treatment time with FAA fixative must be determined to establish uniformity. Statistical analysis details:
Although the use of the new multiple range test by Duncan is appropriate, the presentation in the statistical section without giving adequate details about where it is used and why it is used is a limitation of the paper. The assumptions and parameters considered for the statistical analyses should have been described by the authors.
Presentation of Data:
The tables and figures should be more descriptive. For example, Table 1 needs to have some footnotes explaining what certain abbreviations stand for, such as P and E. Also, some figures lack a scale bar and hence make it hard for one to judge the size proportions.
Discussion Depth:
Discussion: More depth is needed in the interpretation of results. Authors are supposed to discuss the consequence that could arise evolutionarily due to the observed difference in pollen morphologies and should compare more robustly with previous studies.
Poor engagement of broader implications of this research on spinach breeding or conservation strategies.
Outdated References:
No major issues were identified with formatting and style. References cited for statistical methods and analyses are somewhat outdated, and the authors should seek more recent literature to support their statements and methods.
Suggestions for Improvement
Abstract: The abstract should be revised with a focus on key results and implications, rather than providing an abundance of background information.
Introduction: There is a need to reorganize the introduction to create a more progressive narrative and to strengthen the logical continuity of information.
More Transparency in the Methods: It would be nice to have more detail on sample sizes and experimental procedures to enable the reproduction of results.
Discussion: Greater discussion of the implications for genetic diversity and potential applications in breeding would be warranted.
References and Typographical Errors: Review typographical errors and edit the references, making sure that they are updated and relevant.
Conclusion
The present manuscript represents a positive contribution to a better understanding of the morphology of spinach pollen. When the identified shortcomings are addressed and the suggested improvements incorporated, the manuscript will be significantly strengthened and of greater use to its intended audience. With thoughtful revisions, this study has the potential to be a seminal reference in the literature on spinach genetics and taxonomy.
Thanks
Author Response
# Response: Dear reviewer, thank you for approve of our work in this paper. We have read your comments and suggestions carefully, and then revised the manuscript according to each of them. Thank you for your efforts put-in to improve the whole quality of this paper.
- Abstract Clarity:
The abstract should be not so wordy. It should make the findings important more visible, and the abstract should summarize points and results arising rather than stating again the rationale of the study.
# Response: We fully acknowledge the reviewer's suggestions. The section of Abstract has been revised and made them concise and comprehensive in manuscript.
- Typographical Errors:
There are a number of typographical errors throughout the manuscript; for example, "markly" should read "markedly," "plolar" should read "polar." The manuscript needs careful proofreading to eliminate such errors.
# Response: The two mistakes have been corrected, and by rereading the original text to prevent similar mistakes from happening again.
- Structure and Flow:
The introduction is somewhat disjointed in terms of information flow. The authors need to provide background information in a more logical manner; for instance, starting off more generally with spinach before getting into pollen morphology.
# Response: We fully endorse the review expert's insights on the importance of the introduction, we have meticulously organized and rewrote the introduction section of our article to ensure it perfectly meets the above standards (section 1).
- Materials and Methods:
It is essential to add more specificity to the description of the sample collection, specifically mentioning how many plants represent each genotype when pollen was sampled. This is very important to provide reproducibility.
# Response: The detail description on the sample collection was added in the manuscript. Sampling was performed at 10 am during the full spinach bloom. The pollen of nine varieties was used to observe the appearance and morphology of the pollen. 20 pollens for each variety were collected to measure the apparent data. This study selected samples with significant characteristic differences. Specifically, the 9 pollen varieties collected included both cultivated and wild species. The part has been added in section 2.1 of manuscript.
- Here, a detailed explanation of the fixation process for pollen is needed. More precisely, the concentration and treatment time with FAA fixative must be determined to establish uniformity. Statistical analysis details:
# Response: To ensure the effectiveness of subsequent scanning electron microscopy observations of pollen, it is crucial to select the appropriate reagents for fixation. The collected pollen was placed in a 50% ethanol solution of FAA (formaldehyde-acetic acid-alcohol) fixative for immersion for 18 hours. After undergoing a series of processing steps such as dehydration, drying, adhesion and gold sputtering, the pollen was finally subjected to scanning electron microscopy analysis. The part has been added in section 2.2 of manuscript.
- Although the use of the new multiple range test by Duncan is appropriate, the presentation in the statistical section without giving adequate details about where it is used and why it is used is a limitation of the paper. The assumptions and parameters considered for the statistical analyses should have been described by the authors.
# Response: Thank you for the reviewer's valuable comments. We have added more description on statistical sections. The part has been added in section 2.3 of manuscript.
- Presentation of Data:
The tables and figures should be more descriptive. For example, Table 1 needs to have some footnotes explaining what certain abbreviations stand for, such as P and E. Also, some figures lack a scale bar and hence make it hard for one to judge the size proportions.
# Response: Thank you for the reviewer's valuable comments. We have carefully checked the all tables and figures in our manuscript. We add all necessary footnotes in the figures and tables.
- Discussion Depth:
Discussion: More depth is needed in the interpretation of results. Authors are supposed to discuss the consequence that could arise evolutionarily due to the observed difference in pollen morphologies and should compare more robustly with previous studies.
# Response: We fully concur with the reviewer's perspective. We had try our best to enrich the discussion. we added more references support our idea and discussed more depth about our finding compared to previous results, such as on utility of pollen on breeding the use of on in manuscript.
Poor engagement of broader implications of this research on spinach breeding or conservation strategies.
# Response: We agree with the reviewers who, as previously stated, have included breeding-related talk in the discussion.
- Outdated References:
No major issues were identified with formatting and style. References cited for statistical methods and analyses are somewhat outdated, and the authors should seek more recent literature to support their statements and methods.
# Response: We have taken into account the reviewers' comments and incorporated some of the latest literature.

Reviewer 3 Report
Comments and Suggestions for Authors
The manuscript focuses on characterizing spinach varieties based on pollen morphology, thus obtaining differences between varieties based on multivariate analysis. Although the work sounds interesting, it is necessary to make the following observations so that it can be published.
Is “genotypes”, no geneotypes (line 12).
Change “shell” for “testa” (line 38).
Add the voltage used to take images and if it was in high vacuum, this in the Morphology Studies section.
Add the quote of Figure 1 inside the text, for example in line 153.
Add the quote of Figure 2 inside the text.
Put polar, not plolar, line 162.
It is important to place the kilovolts used to take the pollen images in each micrograph because readers can have a clear idea that the pictures were adequate and that there are no aberrations in the pollen morphology.
Within each image of Figure 2, place letters that mark the important points of the pollen morphology, so that, in the description of the figure, they place the meaning of what is being observed, for example, Gr: granules; Q: Pore. I recommend this at least for images 1a, 2a, 3a to 9a.
It is not necessary to report principal component 4, because eigenvalues of 0.000 were obtained, so it is not informative, only showing up to principal component 3 in Table 2.
It remains to justify the separation of the groups through cluster analysis, seek an explanation as to why the wild variety is inserted with domesticated varieties (Figure 3), consider issues such as climatic regions of growth, if it is related to any other structure of the plant, some determining factor such as the type of soil in which they grow, etc. Without this important point, the manuscript is not relevant, it is the crucial part, seeking an explanation for the grouping of the varieties based on their morphological characteristics of the pollen, or adding more variables from the same pollen if it is difficult to explain the groupings.
Author Response
# Response: Dear reviewer, thank you for approve of our work in this paper. We have read your comments and suggestions carefully, and then revised the manuscript according to each of them. During the modification process, adjustments were made to the content of the article, which resulted in changes to the corresponding line numbers. We have since re-annotated these changes in detail to ensure accuracy and consistency.
- Is “genotypes”, no geneotypes (line 12).
# Response: Thank you to the reviewer for pointing out the typographical error, which I have corrected in the text. Additionally, I have reviewed and revised the entire document (line 12).
- Change “shell” for “testa” (line 38).
# Response: This has been corrected in manuscript (line 37).
- Add the quote of Figure 1 inside the text, for example in line 153.
# Response: The content has been supplemented accordingly in the text (line 163).
- Add the quote of Figure 2 inside the text.
# Response: This has been corrected in manuscript (line 186).
- Put polar, not plolar, line 162.
# Response: This has been corrected in manuscript (line 171).
- Add the voltage used to take images and if it was in high vacuum, this in the Morphology Studies section.
It is important to place the kilovolts used to take the pollen images in each micrograph because readers can have a clear idea that the pictures were adequate and that there are no aberrations in the pollen morphology.
# Response: Regarding the two issues mentioned above, the necessary corrections have been made in the manuscript. In morphological studies, the specific parameters of imaging are crucially important. Particularly in images taken using high vacuum conditions and specific voltages, providing these details can greatly enhance readers' understanding of the image quality and accuracy. To ensure the clarity and accuracy of the images, we employed an 8 kV voltage for capturing pollen images under high vacuum conditions. This approach ensures that readers can clearly observe the morphology of the pollen without any distortions caused by improper imaging conditions (line 177).
- Within each image of Figure 2, place letters that mark the important points of the pollen morphology, so that, in the description of the figure, they place the meaning of what is being observed, for example, Gr: granules; Q: Pore. I recommend this at least for images 1a, 2a, 3a to 9a.
# Response: We wholeheartedly agree with your suggestion, as clearly marking key sections helps us grasp and comprehend the information we need to convey more accurately and clearly. The issue has been rectified in the manuscript. To ensure the clarity of the images and to prevent any confusion, we have only added annotations to Figures 2a and 2b. The part has been added on line 177 of manuscript.
- It is not necessary to report principal component 4, because eigenvalues of 0.000 were obtained, so it is not informative, only showing up to principal component 3 in Table 2.
# Response: We have completed the removal of the fourth principal component in Table 2 (line 222).
- It remains to justify the separation of the groups through cluster analysis, seek an explanation as to why the wild variety is inserted with domesticated varieties (Figure 3), consider issues such as climatic regions of growth, if it is related to any other structure of the plant, some determining factor such as the type of soil in which they grow, etc. Without this important point, the manuscript is not relevant, it is the crucial part, seeking an explanation for the grouping of the varieties based on their morphological characteristics of the pollen, or adding more variables from the same pollen if it is difficult to explain the groupings.
# Response: Spinach originated in Iran and is now widely distributed in both China and the United States. Wild spinach is very variable from cultivated spinach. For most crops, there are reproductive isolation between cultivated varieties and wild types. Unlike other wild spinach can crossed with cultivated spinach. Our finding showed there no obvious difference of pollen characteristic between of wild. This maybe lead to the phenomenon.

Round 2
Reviewer 1 Report
Comments and Suggestions for Authors
Upon reviewing the revised version, I appreciate the effort you’ve made to refine the content. However, there are still parts where further improvement is needed to enhance the quality and clarity of the research data.
- Research data: While most sections show improvement, the overall content still requires deeper elaboration:
Accurate measurements can only be achieved in fully hydrated pollen, but the quality of the SEM preparations is poor, since the pollen grains are not fully hydrated, and apertures are sunken. A combined LM and SEM study would have been beneficial, since pollen measurements of fully hydrated pollen in LM are much easier to conduct, and also pollen wall measurements could have been beneficial for the interpretation and conclusions.
Line 137-143: The applied method for measuring pollen is still incorrect as the polar axis and equatorial diameter cannot be measured in isodiametric pollen, instead the pollen diameter should be measured. The P/E ratio is isodiametric.
In the “Material and Methods”, line 132, the authors are describing the “presence of muri”? There are no muri present in the investigated spinach pollen???
I encourage you to re-examine the pollen morphology with LM and SEM, ensuring it aligns more cohesively with the objectives of the study.
- Conclusions: The conclusions presented do not appear to be supported by the data and analyses provided. It is essential to base conclusions on a clear and accurate interpretation of the results. Consider revisiting your data analysis and ensuring that the conclusions you draw are not overreaching or unsupported by the presented research data.

Author Response
Research data: While most sections show improvement, the overall content still requires deeper elaboration:
Accurate measurements can only be achieved in fully hydrated pollen, but the quality of the SEM preparations is poor, since the pollen grains are not fully hydrated, and apertures are sunken. A combined LM and SEM study would have been beneficial, since pollen measurements of fully hydrated pollen in LM are much easier to conduct, and also pollen wall measurements could have been beneficial for the interpretation and conclusions.
# Response: Thank you very much for the valuable comments provided by the reviewer. Please allow me to express my opinion: firstly, the dry state is usually more appropriate if we want to measure the morphological characteristics of the pollen, dry pollen is easier to observe its details and is not deformed by evaporation of water. Scanning electron microscopy (SEM) requires dry pollen samples. Secondly, By using LM, we can observe the internal structure of pollen in hydration state, SEM can reveal the surface texture of pollen surface in dry state. So a combined LM and SEM study would have been beneficial, we will increase the use of LM in future studies.
Line 137-143: The applied method for measuring pollen is still incorrect as the polar axis and equatorial diameter cannot be measured in isodiametric pollen, instead the pollen diameter should be measured. The P/E ratio is isodiametric.
# Response: Thank you very much for helping us identify mistakes. We found that the applied method for measuring pollen in the original text was inappropriate. Because pollen grains of Spinach are isodiametric and spheroidal. For the nearly spherical pollen, what was measured should be called the diameter. And we have modified the language in the manuscript. By referring to the other papers on Amaranthaceae pollen studies for parameters measured or calculated, such as in the study by Angeli et al. (2014). We made observations of other morphological data (the maximum diameter of the pollen and germination pore, the number of visible pores, the density of pore and spinule).
In the “Material and Methods”, line 132, the authors are describing the “presence of muri”? There are no muri present in the investigated spinach pollen???
# Response: We feel very sorry for this misrepresentation in the manuscript, and this error feature has been deleted.
I encourage you to re-examine the pollen morphology with LM and SEM, ensuring it aligns more cohesively with the objectives of the study.
# Response: Thank you very much for this suggestion, using LM combined with SEM will be used in future studies.
Conclusions: The conclusions presented do not appear to be supported by the data and analyses provided. It is essential to base conclusions on a clear and accurate interpretation of the results. Consider revisiting your data analysis and ensuring that the conclusions you draw are not overreaching or unsupported by the presented research data.
# Response: In particular, we thank the reviewers for the detailed comments on the article, and by consulting a palynologist from other universities, we revised the observation method and rearranged the results.

Reviewer 3 Report
Comments and Suggestions for Authors
This version is better, you make the corrections propertly
Author Response
Comments and Suggestions for Authors
This version is better, you make the corrections propertly
# Response: Thank you very much for the valuable comments provided by the reviewer. We will work even harder to make the manuscript universally accepted.
